# A split-GAL4 driver line resource for *Drosophila* neuron types

Geoffrey W Meissner[1]*, Allison Vannan[1], Jennifer Jeter[1], Kari Close[1], Gina M DePasquale[1], Zachary Dorman[1], Kaitlyn Forster[1], Jaye Anne Beringer[1], Theresa Gibney[1], Joanna H Hausenfluck[1], Yisheng He[1], Kristin Henderson[1], Lauren Johnson[1], Rebecca M Johnston[1], Gudrun Ihrke[1], Nirmala A Iyer[1], Rachel Lazarus[1], Kelley Lee[1], Hsing-Hsi Li[1], Hua-Peng Liaw[1], Brian Melton[1], Scott Miller[1], Reeham Motaher[1], Alexandra Novak[1], Omotara Ogundeyi[1], Alyson Petruncio[1], Jacquelyn Price[1], Sophia Protopapas[1], Susana Tae[1], Jennifer Taylor[1], Rebecca Vorimo[1], Brianna Yarbrough[1], Kevin Xiankun Zeng[1], Christopher T Zugates[1], Heather Dionne[1], Claire Angstadt[1], Kelly Ashley[1], Amanda Cavallaro[1], Tam Dang[1], Guillermo A Gonzalez III[1], Karen L Hibbard[1], Cuizhen Huang[1], Jui-Chun Kao[1], Todd Laverty[1], Monti Mercer[1], Brenda Perez[1], Scarlett Rose Pitts[1], Danielle Ruiz[1], Viruthika Vallanadu[1], Grace Zhiyu Zheng[1], Cristian Goina[1], Hideo Otsuna[1], Konrad Rokicki[1], Robert R Svirskas[1], Han SJ Cheong[1], Michael-John Dolan[1], Erica Ehrhardt[1,2], Kai Feng[1,3], Basel El Galfi[1], Jens Goldammer[1,2], Stephen J Huston[1,4], Nan Hu[1], Masayoshi Ito[1], Claire McKellar[1], Ryo Minegishi[1,3], Shigehiro Namiki[1], Aljoscha Nern[1], Catherine E Schretter[1], Gabriella R Sterne[1,5], Lalanti Venkatasubramanian[1], Kaiyu Wang[1], Tanya Wolff[1], Ming Wu[1], Reed George[1], Oz Malkesman[1], Yoshinori Aso[1]*, Gwyneth M Card[1]*, Barry J Dickson[1,3]*, Wyatt Korff[1]*, Kei Ito[1,2]*, James W Truman[1]*, Marta Zlatic[1]*, Gerald M Rubin[1]*, FlyLight Project Team[1]

[1]Janelia Research Campus, Howard Hughes Medical Institute, Ashburn, United States; [2]Institute of Zoology, University of Cologne, Cologne, Germany; [3]Queensland Brain Institute, University of Queensland, Brisbane, Australia; [4]Mortimer B. Zuckerman Mind Brain Behavior Institute, Columbia University, New York, United States; [5]Department of Cell & Molecular Biology, University of California, Berkeley, Berkeley, United States

*For correspondence:
meissnerg@janelia.hhmi.org (GWM);
asoy@janelia.hhmi.org (YA);
cardg@janelia.hhmi.org (GMC);
b.dickson@uq.edu.au (BJD);
korffw@janelia.hhmi.org (WK);
k.ito@uni-koeln.de (KI);
jwt@uw.edu (JWT);
mzlatic@mrc-lmb.cam.ac.uk (MZ);
rubing@janelia.hhmi.org (GMR)

## eLife Assessment

This **valuable** study presents a resource for researchers using *Drosophila* to study neural circuits, in the form of a collection of split-Gal4 lines with an online search engine, which will facilitate the mapping of neuronal circuits. The evidence is **convincing** to demonstrate the utility of these new tools, and of the search engine, for understanding expression patterns in adults and larvae, and differences between the sexes. These resources will be of broad interest to Drosophila researchers in the field of neurobiology.

**Abstract** Techniques that enable precise manipulations of subsets of neurons in the fly central nervous system (CNS) have greatly facilitated our understanding of the neural basis of behavior. Split-GAL4 driver lines allow specific targeting of cell types in *Drosophila melanogaster* and other species. We describe here a collection of 3060 lines targeting a range of cell types in the adult *Drosophila* CNS and 1373 lines characterized in third-instar larvae. These tools enable functional, transcriptomic, and proteomic studies based on precise anatomical targeting. NeuronBridge and

other search tools relate light microscopy images of these split-GAL4 lines to connectomes reconstructed from electron microscopy images. The collections are the result of screening over 77,000 split hemidriver combinations. Previously published and new lines are included, all validated for driver expression and curated for optimal cell-type specificity across diverse cell types. In addition to images and fly stocks for these well-characterized lines, we make available 300,000 new 3D images of other split-GAL4 lines.

## Introduction

The ability to manipulate small subsets of neurons is critical to many of the experimental approaches used to study neuronal circuits. In *Drosophila*, researchers have generated genetic lines that express an exogenous transcription factor, primarily GAL4, in a subset of neurons (*Griffith, 2012*; *Venken et al., 2011*). The GAL4 protein then drives expression of indicator or effector genes carried in a separate UAS transgenic construct (*Fischer et al., 1988*; *Brand and Perrimon, 1993*). This modular approach has proven to be very powerful but depends on generating collections of lines with reproducible GAL4 expression limited to different, specific subsets of cells. In the 1990s, so-called enhancer trap lines were the method of choice (*Bellen et al., 1989*). In this method, a GAL4 gene that lacks its own upstream control elements is inserted as part of a transposable element into different genome locations where its expression might come under the control of nearby endogenous regulatory elements. However, the resultant patterns were generally broad, with expression in hundreds of cells, limiting their use for manipulating specific neuronal cell types (*Manseau et al., 1997*; *Yoshihara and Ito, 2000*; *Ito et al., 2003*).

The FlyLight Project Team (https://www.janelia.org/project-team/flylight) was started to address this limitation with the overall goal of generating a large collection of GAL4 lines that each drove expression in a distinct small subset of neurons—ideally individual cell types. Many labs studying the upstream regulatory elements of individual genes in the 1980s and 1990s had observed that short segments of DNA located upstream of protein-coding regions or in introns, when assayed for enhancer function, frequently drove expression in small, reproducible subsets of cells (see *Levine and Tjian, 2003*). *Pfeiffer et al., 2008* developed an efficient strategy for scaling up such assays of individual DNA fragments and showed that a high percentage of 2–3 kb genomic fragments, when cloned upstream of a core promoter driving GAL4 expression, produced distinct patterns of expression. Importantly, these patterns were much sparser than those observed with enhancer traps (*Pfeiffer et al., 2008*). The approach also took advantage of newly developed methods for site-specific integration of transgenes into the genome (*Groth et al., 2004*), which facilitated the ability to compare constructs by placing them in the same genomic context. FlyLight was established in 2009 to scale up this approach. In 2012, the project reported the expression patterns produced by 6650 different genomic segments in the adult brain and ventral nerve cord (VNC) (*Jenett et al., 2012*) and later in the larval central nervous system (CNS; *Li et al., 2014*). In the adult central brain, we estimated that this 'Generation 1' (Gen1) collection contained 3850 lines in which the number of labeled central brain neurons was in the range of 20–5000. These GAL4 driver lines and a collection of similarly constructed LexA lines have been widely used by hundreds of laboratories. However, we concluded that less than 1% of our lines had expression in only a single cell type, highlighting the need for a better approach to generating cell-type-specific driver lines.

To gain more specific expression the project turned to intersectional methods. These methods require two different enhancers to be active in a cell to observe expression of a functional GAL4 transcriptional activator in that cell. We adopted the split-GAL4 approach that was developed by *Luan et al., 2006* and subsequently optimized by *Pfeiffer et al., 2010* in which an enhancer drives either the activation domain (AD) or the DNA-binding domain (DBD) of GAL4 (or optimized alternatives) in separate proteins. When present in the same cell the proteins carrying the AD and DBD domains, each inactive in isolation, dimerize to form a functional GAL4 transcription factor.

The work of *Jenett et al., 2012* described the expression patterns of thousands of enhancers. Using these data, anatomical experts could identify Gen1 enhancers that express in the cell type of interest and cross flies that express the AD or DBD half of GAL4 under the control of two of the same enhancers and observe the resultant intersected expression pattern. Large collections of such AD or DBD genetic drivers, which we refer to as hemidriver lines, were generated at Janelia (*Dionne et al.,*

*2018*) and at the IMP in Vienna (*Tirian and Dickson, 2017*). In ~5–10% of such crosses, the cell type of interest was still observed, but now as part of a much sparser expression pattern than displayed by either of the initial enhancers. In about 1–2% of these genetic intersections, expression appeared to be limited to a single cell type.

At Janelia, early efforts were directed at targeting neuronal populations in the optic lobes (*Tuthill et al., 2013*; *Nern et al., 2015*; *Wu et al., 2016*) and mushroom bodies (*Aso et al., 2014*). We were encouraged by the fact that we were able to generate lines specific for the majority of cell types in these populations. With the involvement of additional collaborating groups, the project was extended to several other CNS regions (*Table 1*). In our initial studies, we relied on expert human annotators performing extensive visual surveys of expression data to identify candidate enhancers to intersect. More recently, two advances have greatly facilitated this process. First, we have developed computational approaches (*Otsuna et al., 2018*; *Hirsch et al., 2020*, *Mais et al., 2021*; *Meissner et al., 2023*) to search databases of neuronal morphologies generated by stochastic labeling (*Nern et al., 2015*) of several thousand of the *Jenett et al., 2012* and *Tirian and Dickson, 2017* GAL4 lines. Second, electron microscopy (EM) datasets (*Scheffer et al., 2020*; *Cheong et al., 2023*; *Marin et al., 2023*; *Takemura et al., 2023*; *Nern et al., 2024*; *Schlegel et al., 2024*) have provided comprehensive cell-type inventories for many brain regions.

In this report, we summarize the results obtained over the past decade and present a collection of the best cell-type-specific split-GAL4 lines that were identified.

## Results
### Cell-type-specific split-GAL4 line collection

We describe here a collection of 3060 split-GAL4 lines targeting cell types across the adult *Drosophila* CNS. All lines were created in collaborations with the FlyLight Project Team at Janelia Research Campus from 2013 to 2023, based on examination of over 77,000 split combinations. 1644 lines were published previously and are drawn together here as part of the larger collection. *Table 1* summarizes prior publications. The remaining 1416 lines in the collection are described in other in preparation manuscripts or newly reported here as shown in *Figure 1—source data 1*, which specifies proper citations and other line-specific information.

To confirm and consistently document the expression patterns of included lines, all were rescreened by crossing to a UAS-CsChrimson-mVenus reporter inserted in at a specific genomic location (attP18) (*Figure 1*; *Klapoetke et al., 2014*). At least one male and one female CNS were dissected, antibody labeled, and imaged per line. Expression patterns were validated by manual qualitative comparison to prior data, where available, and scored for specificity and consistency (see below and Methods). Male/female differences were observed in 109 of these 3060 lines, confirming previously reported sexual dimorphisms or suggesting potential areas for further study (*Figure 1F*; *Figure 1—source data 1*).

Previously published and newer lines were evaluated as a whole, taking advantage of new screening data to highlight existing lines or identify new lines that best label cell types. The lines were selected for inclusion based on several factors:

1. Diversity: Where cell-type information was available, especially from comparison to EM volumes (*Scheffer et al., 2020*; *Cheong et al., 2023*; *Marin et al., 2023*; *Takemura et al., 2023*; *Nern et al., 2024*), we generally limited each cell type to the two best split-GAL4 lines, so as to cover a wider range of cell types.
2. Specificity: 1767 lines were scored as highest quality, well suited to activation-based behavioral studies, with strong and consistent labeling of a single identified cell type and minimal detected off-target expression (*Figure 2A*; *Figure 1—source data 1*). 1258 lines showed spatially segregated off-target expression that does not interfere with neuron visualization for anatomy or physiology (*Figure 2B, C*; *Figure 1—source data 1*). A control line with minimal detected expression was also included (SS01062; *Namiki et al., 2018*).
3. Consistency: 34 lines were very specific but showed weaker or less consistent expression (*Figure 2D*; *Figure 2—figure supplement 1*; *Figure 1—source data 1*). These lines reveal anatomy but may be challenging to use for manipulations without examination of expression in each individual fly.
4. Regions of interest: Lines were generated in collaboration with Janelia labs and collaborators studying particular CNS regions or classes of neurons. While this collection includes many

**Table 1.** Publications reporting split-GAL4 lines from the adult cell-type-specific collection.

Publications are listed by year. The number of lines from the collection in each publication is listed, along with the central nervous system (CNS) regions and/or cell types most commonly labeled. Many of these publications describe additional lines that were not included in the collection described here.

| Publication DOI | First author(s) | Year | Citation | Split-GAL4 lines | Anatomical region/cell types for lines |
|---|---|---|---|---|---|
| 10.1016/j.neuron.2013.05.024 | Tuthill, Nern | 2013 | *Tuthill et al., 2013* | 22 | Lamina (optic lobe) |
| 10.1016/j.neuron.2014.05.017 | Feng, Palfreyman | 2014 | *Feng et al., 2014* | 1 | SAG ascending neurons |
| 10.7554/eLife.04577 | Aso | 2014 | *Aso et al., 2014* | 63 | Mushroom body |
| 10.7554/eLife.16135 | Aso | 2016 | *Aso and Rubin, 2016* | 2 | Mushroom body |
| 10.7554/eLife.21022 | Wu, Nern | 2016 | *Wu et al., 2016* | 56 | Lobula columnar neurons (optic lobe) |
| 10.1016/j.neuron.2017.03.010 | Strother | 2017 | *Strother et al., 2017* | 9 | T4 neurons and inputs (optic lobe) |
| 10.1016/j.neuron.2017.05.036 | von Reyn | 2017 | *von Reyn et al., 2017* | 1 | Lobula columnar neuron LC4 (optic lobe) |
| 10.1038/nature24626 | Klapoetke | 2017 | *Klapoetke et al., 2017* | 3 | LPLC2 neurons and inputs (optic lobe) |
| 10.7554/eLife.24394 | Takemura | 2017 | *Takemura et al., 2017* | 1 | CT1 neurons (optic lobe) |
| 10.1002/cne.24512 | Wolff | 2018 | *Wolff and Rubin, 2018* | 46 | Central complex |
| 10.7554/eLife.34272 | Namiki | 2018 | *Namiki et al., 2018* | 137 | Descending neurons |
| 10.1016/j.cub.2019.01.009 | Jovanic | 2019 | *Jovanic et al., 2019* | 2 | Larval anemotaxis and adult descending neuron |
| 10.7554/eLife.43079 | Dolan | 2019 | *Dolan et al., 2019* | 2 | Lateral horn |
| 10.1016/j.cub.2020.07.083 | Wang,Wang | 2020 | *Wang et al., 2020a* | 5 | Descending neurons DNp13 |
| 10.1016/j.neuron.2020.08.006 | Turner-Evans | 2020 | *Turner-Evans et al., 2020* | 2 | Central complex |
| 10.1038/s41467-020-19936-x | Feng | 2020 | *Feng et al., 2020* | 80 | Leg motor MDN targets |
| 10.1038/s41586-020-2055-9 | Wang, Wang | 2020 | *Wang et al., 2020b* | 9 | Descending neurons oviDN |
| 10.1371/journal.pone.0236495 | Bogovic | 2020 | *Bogovic et al., 2020* | 1 | Brain |
| 10.7554/eLife.50901 | Davis, Nern | 2020 | *Davis et al., 2020* | 40 | Optic lobe |
| 10.7554/eLife.57685 | Morimoto | 2020 | *Morimoto et al., 2020* | 10 | Central brain targets of lobula LC6 neurons |
| 10.7554/eLife.58942 | Schretter | 2020 | *Schretter et al., 2020* | 12 | Central brain pC1d aIPg |
| 10.1038/s41586-020-2972-7 | Wang1, Wang | 2021 | *Wang et al., 2021* | 7 | Descending neurons vpoDN vpoEN |
| 10.1101/2021.07.23.453511 | Mais | 2021 | *Mais et al., 2021* | 1 | Central brain pC1e |
| 10.7554/eLife.66039 | Hulse, Haberkern, Franconville, Turner-Evans | 2021 | *Hulse et al., 2021* | 1 | Central complex |
| 10.7554/eLife.71679 | Sterne | 2021 | *Sterne et al., 2021* | 67 | Subesophageal zone |
| 10.7554/eLife.71858 | Kind, Longden, Nern, Zhao | 2021 | *Kind et al., 2021* | 3 | R7 and R8 photoreceptor targets (optic lobe) |
| 10.1016/j.cub.2022.01.008 | Namiki, Ros | 2022 | *Namiki et al., 2022* | 9 | Descending neurons flight |
| 10.1016/j.cub.2022.06.019 | Baker | 2022 | *Baker et al., 2022* | 28 | Central brain AMMC, WED, AVLP, and PVLP |
| 10.1016/j.neuron.2022.02.013 | Klapoetke | 2022 | *Klapoetke et al., 2022* | 2 | Lobula LC18 and LC25 neurons (optic lobe) |
| 10.1101/2022.12.14.520178 | Zhao | 2022 | *Zhao et al., 2022* | 1 | H2 neurons (optic lobe) |
| 10.1038/s41586-023-06271-6 | Vijayan | 2023 | *Vijayan et al., 2023* | 2 | Descending neurons oviDN |
| 10.1101/2023.05.31.542897 | Ehrhardt, Whitehead | 2023 | *Ehrhardt et al., 2023* | 164 | Dorsal VNC |
| 10.1101/2023.06.07.543976 | Cheong, Eichler, Stuerner | 2023 | *Cheong et al., 2023* | 11 | VNC premotor |
| 10.1101/2023.06.21.546024 | Isaacson | 2023 | *Isaacson et al., 2023* | 6 | Lobula plate LPC and LLPC (optic lobe) |
| 10.1101/2023.10.16.562634 | Zhao | 2023 | *Zhao et al., 2023* | 2 | MeLp2 and LPi4b neurons (optic lobe) |
| 10.1101/2023.11.29.569241 | Garner, Kind | 2023 | *Garner et al., 2023* | 5 | Medulla to AOTU (MeTu) neurons (optic lobe) |
| 10.25378/janelia.23726103 | Minegishi | 2023 | *Minegishi et al., 2023* | 52 | Ascending neurons (see also *Chen et al., 2023a*) |

*Table 1 continued on next page*

*Table 1 continued*

| Publication DOI | First author(s) | Year | Citation | Split-GAL4 lines | Anatomical region/cell types for lines |
|---|---|---|---|---|---|
| 10.1038/s41467-023-43566-8 | Longden | 2023 | *Longden et al., 2023* | 1 | L1 neurons (optic lobe) |
| 10.1016/j.cub.2024.01.015 | Lillvis | 2024 | *Lillvis et al., 2024* | 22 | VNC song generation |
| 10.1016/j.cub.2024.01.071 | Cheong, Boone, Bennett | 2024 | *Cheong et al., 2024* | 1 | Ascending neurons flight |
| 10.1038/s41586-024-07222-5 | Gorko | 2024 | *Gorko et al., 2024* | 3 | Neck motor neurons |
| 10.1101/2024.03.15.585289 | Schretter | 2024 | *Schretter et al., 2024* | 3 | Aggression circuit |
| 10.1101/2024.04.16.589741 | Nern | 2024 | *Nern et al., 2024* | 320 | Optic lobe |
| 10.1101/2024.10.21.619448 | Wolff | 2024 | *Wolff et al., 2024* | 240 | Central complex |
| 10.7554/eLife.90523 | Rubin | 2024 | *Rubin and Aso, 2024* | 21 | Mushroom body output neurons |
| 10.7554/eLife.94168.3 | Shuai | 2024 | *Shuai et al., 2024* | 168 | Mushroom body |
| Dionne et al., in prep | Dionne | | | 81 | Accessory medulla (optic lobe) and clock |
| Rubin et al., in prep | Rubin | | | 50 | Anterior optic tubercle |
| This paper | | | | 1285 | |
| Total | | | | 3060 | |

high-quality lines across the CNS, most efforts were directed to regions of interest within the CNS (see below and *Table 1*).

We examined the distribution of the cell-type collection across the male and female CNS. To visualize the distribution, we aligned each image to a unisex template and segmented neuron patterns from rescreening images into a heat map (*Figure 3*). The cell-type lines show neuronal expression across most of the CNS, with 96% of voxels having expression from at least 5 lines, 90% with 10 or more lines, and 66% with 20 or more lines (*Figure 3—source data 1*). Prominently labeled areas include the fan-shaped body, lobula, and superior medial protocerebrum in the brain, and the T1 medial ventral association center and intermediate tectulum in the VNC. The antennal lobe is very rarely labeled. Other relatively rarely labeled areas include the anterior ventrolateral protocerebrum and prow in the brain and the ovoid (accessory mesothoracic neuropil) in the VNC.

We compared the line distribution between female and male images (*Figure 3C, D*). We observed more female split-GAL4 expression in the epaulette and abdominal ganglion, and more male expression in the antennal mechanosensory and motor center and several regions of the mushroom body (*Figure 3—source data 1*). Two example sexually dimorphic split-GAL4 lines from the cell-type collection illustrate regions of increased split-GAL4 expression in females (*Figure 3E–H*). The lines were identified based on NeuronBridge searches of the 'female minus male' image (*Clements et al., 2021*; *Clements et al., 2024*). Examination of individual male images suggests that (in contrast to the female) the subtraction approach highlighted male artifacts of the reporter system: either background incompletely segmented away from neurons (e.g. in the metathoracic ganglion) or repetitive neuron signal unrelated to other line-specific expression ('sparse-T' neuron and projections into the antennal mechanosensory and motor center; *Meissner et al., 2023*).

The collection of lines described above has been deposited with Bloomington *Drosophila* Stock Center (Bloomington, IN) for availability until at least August 31, 2026. Images and line metadata are available at https://splitgal4.janelia.org. Anatomical searching for comparison to other light microscopy (LM) and EM data is available at https://neuronbridge.janelia.org.

## Raw image data collection and analysis workflow

In addition to the line collection described above, we describe a split-GAL4 image data resource. It consists of all good quality FlyLight split-GAL4 images and associated metadata generated in the project's first 9 years of split-GAL4 characterization (*Figure 1—source data 3*). It includes additional data for the lines described above, together with data for many additional split-Gal4 combinations. Many of these represent lower quality lines that label multiple nearby cell types (*Figure 2D* and see below), but the image collection also includes additional high-quality lines that were not chosen for

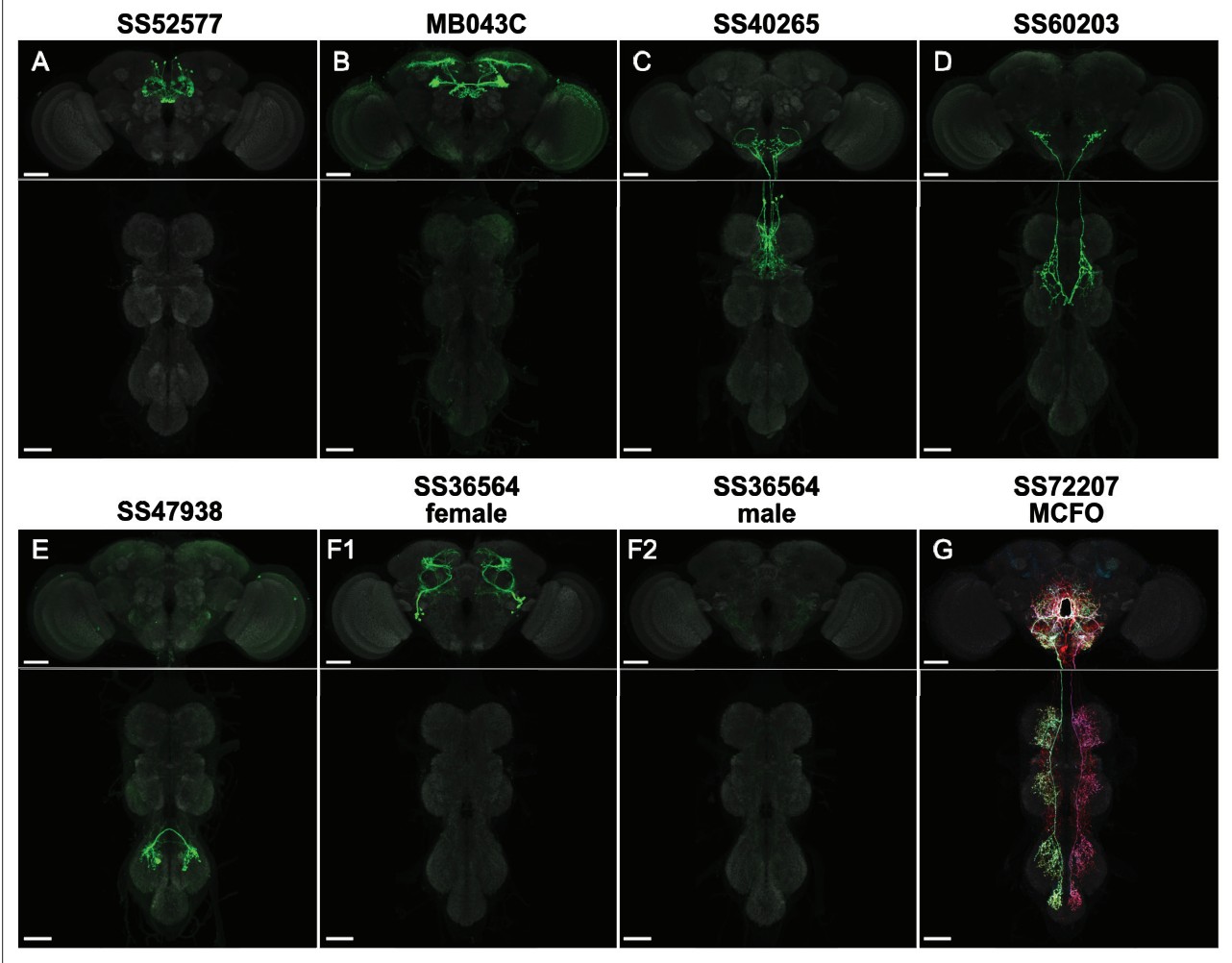

**Figure 1.** Example cell-type-specific lines. (**A**) Split-GAL4 line SS52577 is expressed in P-FNv neurons arborizing in the protocerebral bridge, fan-shaped body, and nodulus (**Wolff and Rubin, 2018**). (**B**) Split-GAL4 line MB043C is expressed primarily in PAM-α1 dopaminergic neurons that mediate reinforcement signals of nutritional value to induce stable olfactory memory for driving wind-directed locomotion and higher-order learning (**Aso et al., 2014**; **Ichinose et al., 2015**; **Aso and Rubin, 2016**; **Aso et al., 2023**; **Yamada et al., 2023**). (**C**) Split-GAL4 line SS40265 is expressed in members of the 8B(t1) cluster of cholinergic neurons that connect the lower tectulum neuropil of the prothorax with the gnathal neuropil and the ventral most border of the vest neuropil of the brain ventral complex. (**D**) Split-GAL4 line SS60203 is expressed in ascending neurons likely innervating the wing neuropil. (**E**) Split-GAL4 line SS47938 is expressed in LBL40, mediating backwards walking (**Feng et al., 2020**; same sample used in Figure 5b, CC-BY license). (**F**) Split-GAL4 line SS36564 is expressed in female-specific aIPg neurons (**F1**) and not observed in males (**F2**; **Schretter et al., 2020**). (**G**) MCFO of split-GAL4 line SS72207 with specific expression in DNg34, a cell type described in **Namiki et al., 2018**. Scale bars, 50 μm. See **Figure 1—source data 1** for more line information and **Supplementary file 1** for images of all cell-type-specific lines.

The online version of this article includes the following source data for figure 1:

**Source data 1.** Spreadsheet of cell-type-specific adult and larval split-GAL4 lines.

**Source data 2.** Spreadsheet of metadata for rescreened cell-type-specific adult split-GAL4 lines.

**Source data 3.** Spreadsheet of image metadata for raw data release.

stabilization or are currently not maintained as stable lines (e.g. because of similarity to other lines in the collection).

FlyLight's analysis workflow consisted of multiple related image-generating pipelines (**Figure 4A, B**). Typically, we screened an 'Initial Split' (IS) cross temporarily combining the two candidate split hemidrivers and reporter before building (stabilizing) a genetically stable 'Stable Split' (SS) line. IS and SS data with the same five to six digit code reflect the same combination of split hemidrivers. Some split-GAL4 lines use region-specific nomenclatures, with names prefixed by 'MB' (mushroom body), 'OL' (optic lobe), or 'LH' (lateral horn), and in most cases label neurons within or nearby those regions.

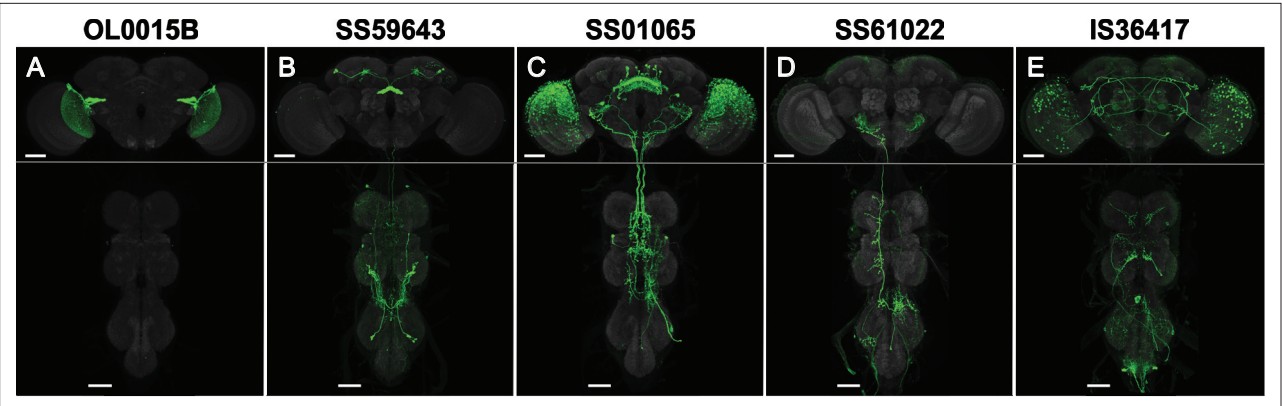

**Figure 2.** Examples of line quality levels. The 3060 cell-type lines were scored for expression. (**A**) Quality level 1 (1767 lines): Split-GAL4 line OL0015B (***Wu et al., 2016***) is specifically and strongly expressed in a single cell type. Occasional weak expression may be seen in other cells. (**B**) Quality level 2 (1232 lines): Split-GAL4 line SS59643 (***Wolff et al., 2024***) has expression in two cell types. Occasional weak expression may be seen in other cells. (**C**) Quality level 3 (26 lines): Split-GAL4 line SS59643 (***Wolff et al., 2024***) has expression in three or more cell types. (**D**) Quality level 4 (34 lines): Split-GAL4 line SS61022 has specific expression but weak or variable labeling efficiency. See ***Figure 2—figure supplement 1*** for examples of variable expression. (**E**) Quality level 5: IS36417 is an Initial Split combination not selected for stabilization. Groups of neurons are visible, but the cell type of interest was not labeled with sufficient specificity for further work. Such lines were only included in the raw image collection. Scale bars, 50 μm.

The online version of this article includes the following figure supplement(s) for figure 2:

**Figure supplement 1.** Example expression variability in Quality level 4 lines.

We examined about 77,000 IS crosses and after quick visual evaluation chose to image a fly CNS sample from about half of them. 36,033 such IS samples (individual flies) are included in this image collection (***Table 2***). About 10% of IS crosses were made into SS lines and imaged again, with 8273 such lines included.

Further documentation of the full SS pattern at higher quality with varying levels of pre-synaptic labeling was generated by the Polarity pipeline (***Aso et al., 2014***; ***Sterne et al., 2021***), followed by full 20× imaging and selected regions of interest imaging at 63×, with 37,409 such samples from 7039 lines included. Single-neuron stochastic labeling by MultiColor FlpOut (MCFO; ***Nern et al., 2015***) revealed single-neuron morphology and any diversity latent within the full SS pattern, with 54,807 such samples from 7679 lines included in the image collection.

In total the raw image collection consists of 46,653 IS/SS combinations, 129,665 samples (flies), 612,124 3D image stacks, and 4,371,364 secondary (processed) image outputs, together 192 TB in size. The IS data may be particularly valuable for work on understudied CNS regions, as it contains biological intersectional results (as opposed to computational predictions) that may have specific expression outside regions focused on in prior publications (***Table 1***).

Due to the size of the raw image collection, it has not undergone the same level of validation as our other image collections, and caution is recommended in interpreting the data therein. We believe it is nonetheless of overall good quality. The images are available at https://flylight-raw.janelia.org and are gradually being made searchable via NeuronBridge due to the size of the dataset. At the time of this version of record, the 36,000 IS images have been added, with the remaining data planned to be added by the end of 2024. For a further guide to interpreting the image data, see ***Supplementary file 2***.

## Larval split-GAL4 line collection

We include 982 split-GAL4 lines, 31 Generation 1 GAL4 lines, and 353 Generation 1 LexA lines selected for specific expression in the third-instar larval CNS (see ***Figure 1—source data 1***). Seven more split-GAL4 lines were selected based on both larval and adult expression.

We selected larval lines based on the following criteria: (1) the projection pattern had to be sparse enough for individual neurons in a line to be identifiable; (2) the line had to be useful for imaging of neural activity or patch-clamp recording; and (3) the line had to be useful for establishing necessity or sufficiency of those neurons in behavioral paradigms. We thus selected lines with no more than 2 neurons per brain hemisphere or per subesophageal zone (SEZ) or VNC hemisegment. When there

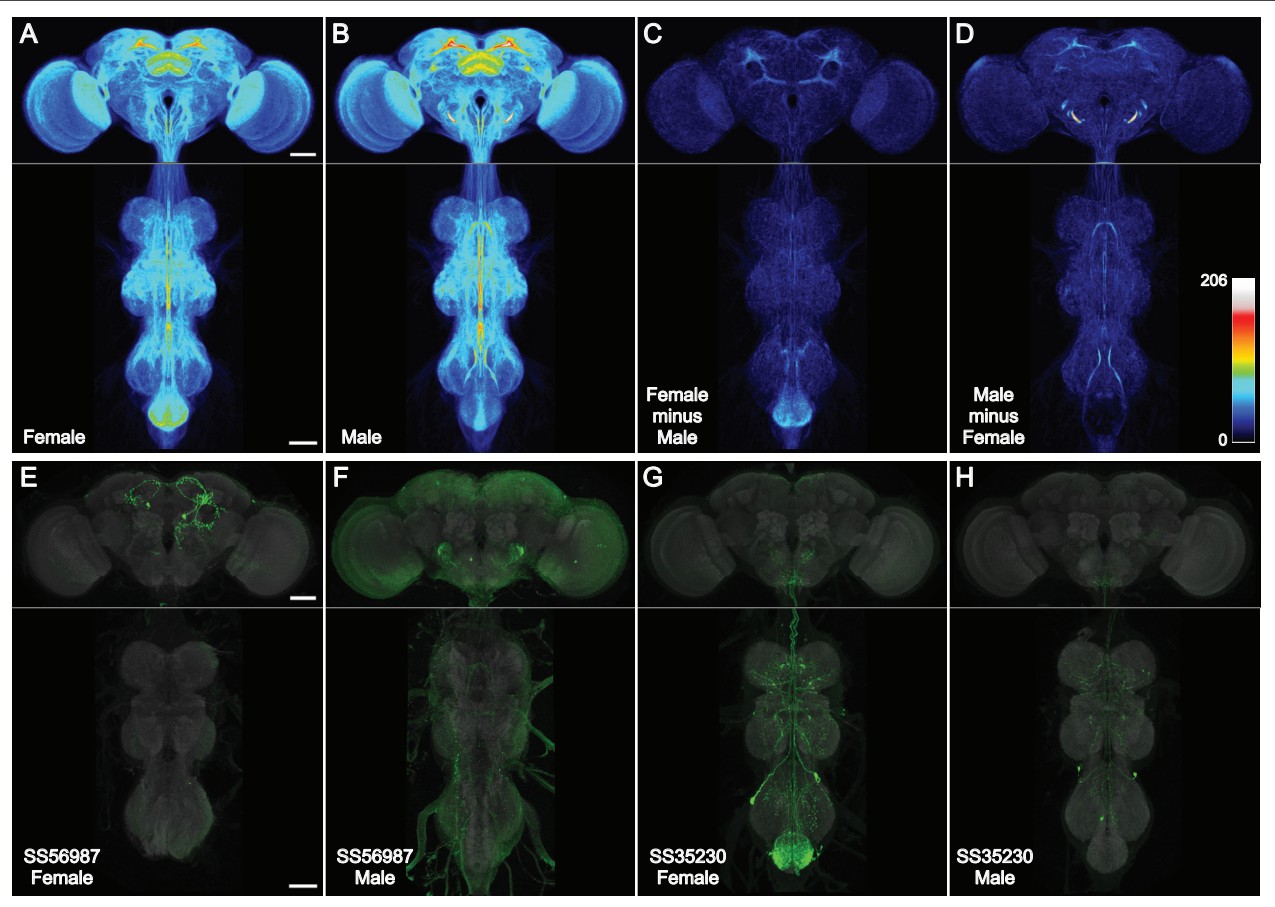

**Figure 3.** Spatial distribution of cell-type lines. (**A–D**) Images of one male and one female sample from 3029 cell-type rescreening lines were aligned to JRC2018 Unisex (**Bogovic et al., 2020**), segmented from background (see Methods), binarized, overlaid, and maximum intensity projected, such that brightness indicates the number of lines with expression. All images were scaled uniformly to a maximum brightness equal to 206 lines on Fiji's 'royal' LUT (scale inset in D). This saturated a small portion of the male antennal mechanosensory and motor center (AMMC) that reached a peak value of 260 lines per voxel, for the purpose of better visualizing the rest of the central nervous system (CNS). (**A**) Female CNS. (**B**) Male CNS. (**C**) Female image stack minus male, then maximum intensity projected. (**D**) Male image stack minus female, then maximum intensity projected. (**E, F**) Split-GAL4 line SS56987 (**Schretter et al., 2020**) is sex-specifically expressed in pC1d neurons that largely lie within the region of the female central brain highlighted in (**C**). Male and female images have different brightness scales. (**G, H**) Split-GAL4 line SS35230 (**Shuai et al., 2024**) is sex-specifically expressed in abdominal ganglion neurons that largely lie within the region of the female ventral nerve cord (VNC) highlighted in (**C**). Male and female images have different brightness scales. All scale bars, 50 µm.

The online version of this article includes the following source data for figure 3:

**Source data 1.** Spreadsheet for coverage analysis.

are more neurons in an expression pattern, overlap in projections makes it hard to uniquely identify these neurons. Also, interpreting neural manipulation results in behavioral experiments becomes much more challenging.

If lines had sparse expression in the brain or SEZ but also had expression in the VNC (sparse or not), we nevertheless considered them useful for studying the brain or SEZ neurons because (1) the brain neurons could be identified; (2) the line could be used for imaging or patch-clamp recording in the brain; and (3) the split-GAL4 line could be cleaned up for behavioral studies, using teashirt-killer-zipper to eliminate VNC expression (**Dolan et al., 2017**). Similarly, lines with specific VNC expression but with undesired additional expression in the brain or SEZ were still useful for imaging and behavioral studies in the VNC, because a loss of phenotype following teashirt-killer-zipper could establish causal involvement of VNC neurons. For LexA lines, we included a few with 3 or more neurons per hemisphere or per hemisegment because there are fewer LexA lines overall and they could still be used for functional connectivity studies.

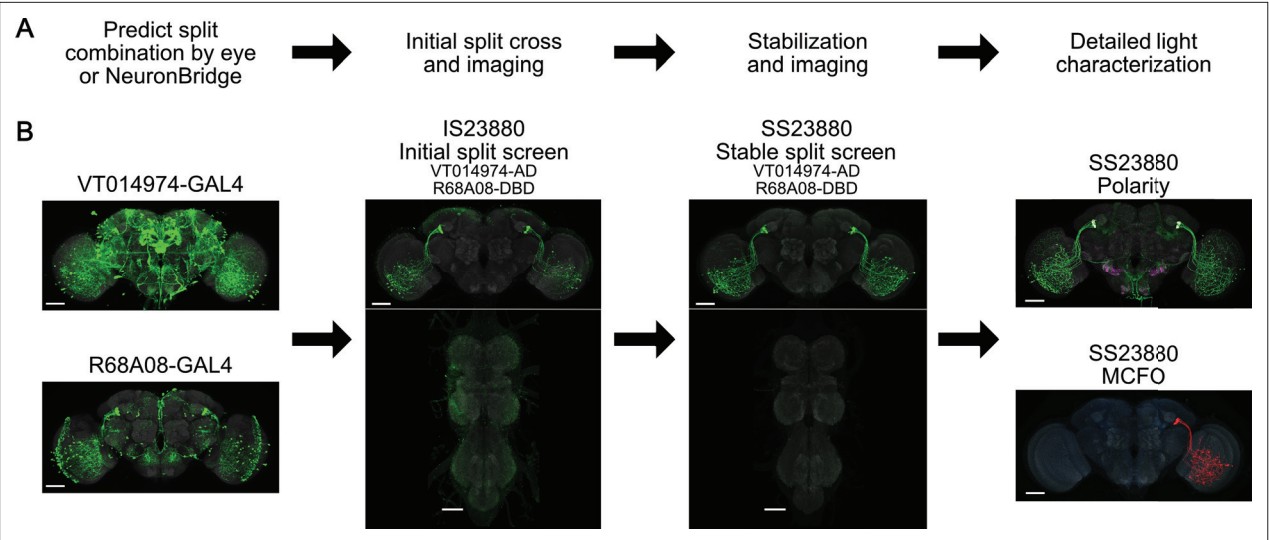

**Figure 4.** Split-GAL4 workflow and FlyLight data release statistics. (**A**) Typical workflow of predicting and characterizing split-GAL4 combinations. (**B**) Example with split-GAL4 line SS23880 (*Garner et al., 2023*).

Images of all of the selected lines are available at Virtual Fly Brain (VFB; https://raw.larval.flylight. virtualflybrain.org/) and will be further integrated into the main VFB website (https://virtualflybrain. org/). We anticipate these lines will be useful tools in conjunction with EM volumes of the larval nervous system.

## Discussion

This collection of cell-type-specific split-GAL4 lines and associated image resource is a toolkit for studies of many *Drosophila* neurons. The adult cell-type-specific line collection covers many identified cell types, and the images are being made instantly searchable for LM and EM comparisons at NeuronBridge, enabling selection of existing or new combinations of genetic tools for specific targeting based on anatomy. In total, the Janelia FlyLight Project Team has contributed 540,000 3D

**Table 2.** Summary of FlyLight adult fly line and image releases.

Includes descriptions of *Jenett et al., 2012*; *Dionne et al., 2018*; *Tirian and Dickson, 2017*; publications in *Table 1*; and stock distribution by Bloomington *Drosophila* Stock Center and Janelia Fly Facility. Section with italic text is specific to publications from *Table 1* and this publication. Image counts are unique between categories, whereas line counts overlap extensively between categories. 'Image tiles' are considered as unique 3D regions, with each MCFO tile captured using two LSM image stacks. Stock shipments count each shipment of each stock separately. Raw and processed images are available at https://www.janelia.org/gal4-gen1, https://gen1mcfo.janelia.org, https://splitgal4.janelia.org, and many can be searched based on anatomy at https://neuronbridge.janelia.org.

| | Gen1 GAL4 | Gen1 LexA | Gen1 MCFO | AD/DBD | IS screen | SS screen | SS MCFO | SS polarity | Total |
|---|---|---|---|---|---|---|---|---|---|
| Lines | 9389 | 1500 | 5155 | 8882 | *35,708* | *8986* | *7823* | *7050* | 67,562 |
| Samples imaged | 9389 | 1500 | 74,363 | NA | *36,182* | *16,974* | *55,702* | *37,526* | 231,636 |
| 20× image tiles | 18,891 | 2966 | 29,780 | NA | *72,346* | *33,924* | *66,484* | *61,787* | 286,178 |
| 40× image tiles | 0 | 0 | 111,563 | NA | *0* | *0* | *0* | *0* | 111,563 |
| 63× image tiles | 0 | 0 | 22,372 | NA | *0* | *0* | *77,039* | *49,117* | 148,528 |
| Samples with 63× images | 0 | 0 | 8489 | NA | *0* | *0* | *31,849* | *17,619* | 57,957 |
| Stocks currently available | 2871 | 1781 | NA | 8358 | *NA* | *3013* | *NA* | *NA* | 16,023 |
| Janelia & Bloomington stock shipments as of September 2024 | 167,082 | 29,227 | NA | 69,546 | NA | *40,067* | NA | NA | 305,922 |

images of 230,000 GAL4, LexA, and split-GAL4 fly CNS samples. As of this publication, Janelia and Bloomington *Drosophila* Stock Center have distributed FlyLight stocks 300,000 times to thousands of groups in over 50 countries. We have also released standardized protocols for fly dissection and immunolabeling at the FlyLight protocols website https://www.janelia.org/project-team/flylight/protocols.

Several caveats should be kept in mind when using these tools and other transgenic systems. Many split-GAL4 lines still have some off-target expression (see *Figure 2*). It is always good practice to validate results using multiple lines labeling the same neurons. In most cases, the off-target expression will not be present in multiple lines, supporting assigning a phenotype to the shared neurons. The less precise a set of lines, the more lines and other supporting evidence should be used. Different effectors and genomic insertion sites also vary in strength and specificity of expression and should be validated for each application (*Pfeiffer et al., 2010*; *Aso et al., 2014*). For example, MCFO images can have brighter single neurons than in full split-GAL4 patterns. UAS-CsChrimson-mVenus allows for a direct correlation between labeling and manipulation that isn't available for every effector. See *Supplementary file 2* for other examples of differences between effectors used in this study. If multiple transgenes are inserted into the same genomic locus, they can interact directly in unanticipated ways (transvection; *Mellert and Truman, 2012*). Fly age should also be controlled, as expression can vary over fly development (*Harris et al., 2015*), and effectors may take time to mature after expression (*Baird et al., 2000*).

Cell type consensus naturally evolves as new observational modalities are developed (e.g. connectomics and RNA sequencing) and comparisons are made across existing data, such as between EM volumes. Not all lines in this collection have cell-type information, and some types will likely change. Different cell types also vary widely in their number of constituent neurons, and within a type there is often some variation between hemispheres and individuals. Thus, for any cell types with more than one neuron we cannot claim to have fully labeled every neuron of a type.

Despite extensive efforts, we have not developed specific lines for every cell type in the fly CNS. Moreover, the peripheral nervous system and the rest of the body were beyond the scope of this effort. Our hemidrivers likely do not effectively label some CNS cell types (discounting extremely broad expression), as they are based on Generation 1 collections that together may only label about half of all biological enhancers. As a rule of thumb, we estimate being able to use these tools to create precise split-GAL4 lines for one-third of cell types, imprecise lines for another third, and no usable line for the remaining third of cell types. This hit rate nonetheless is often enough to characterize key circuit components. Continued iteration with the existing toolset is expected to lead to diminishing returns on improved CNS coverage.

An ideal toolkit with complete coverage of cell types with specific lines would require additional development. Capturing a full range of enhancers requires alternatives to our model of short genetic fragments inserted into a small number of genomic locations. Predicting enhancer combinations based on transcriptomic data and capturing them at their native location holds promise (e.g. *Pavlou et al., 2016*; *Chen et al., 2023b*), though high-quality transcriptomics data with cell-type resolution are not yet available for much of the CNS. Insertions throughout the genome bring their own challenges of unpredictable side-effects, however, and specific targeting of every cell type remains a challenge. Further restriction by killer zipper, GAL80, Flp, or other intersectional techniques will likely remain an area of development (*Pavlou et al., 2016*; *Dolan et al., 2017*; *Ewen-Campen et al., 2023*).

The Janelia FlyLight Project Team has achieved its goal of developing tools to study the *Drosophila* nervous system. The team and collaborators have delivered GAL4/LexA lines labeling many neurons, single-neuron images enabling correlation with EM and the prediction of split-GAL4 lines, this collection of cell-type-specific split-GAL4 lines, an image resource of many more split combinations, and a series of focused studies of the nervous system.

## Materials and methods

**Key resources table**

| Reagent type (species) or resource | Designation | Source or reference | Identifiers | Additional information |
|---|---|---|---|---|
| Genetic reagent (*Drosophila melanogaster*) | MCFO-1; hsPESTOPT_attP3_ 3stop1_X_0036; (w, pBPhsFlp2::PEST in attP3;; pJFRC201- 10XUASFRT >STOP > FRT-myr::smGFP-HA in VK00005,pJFRC240- 10XUAS-FRT>STOP > FRT- myr::smGFP-V5-THS-10XUASFRT >STOP > FRT- myr::smGFPFLAG in su(Hw)attP1/TM3,Sb) | *Nern et al., 2015* | RRID:BDSC_64085 (Janelia stock 1117734) | |
| Genetic reagent (*Drosophila melanogaster*) | MCFO-2; pBPhsFLP_PEST_ HAV5_FLAG_OLLAS_ X3_0095; (w, pBPhsFlp2::PEST in attP3;; pJFRC210- 10XUASFRT >STOP > FRT-myr::smGFP-OLLAS in attP2, pJFRC201- 10XUAS-FRT>STOP > FRT- myr::smGFP-HA in VK0005, pJFRC240-10XUAS- FRT>STOP > FRT-myr::smGFP-V5-THS10XUAS- FRT>STOP > FRT-myr::smGFPFLAG in su(Hw)attP1/TM2) | *Nern et al., 2015* | RRID:BDSC_64086 (Janelia stock 3022015) | |
| Genetic reagent (*Drosophila melanogaster*) | MCFO-4; 57C10wt_attp8_ 3stop1; (w, R57C10-Flp2 in su(Hw)attP8;; pJFRC201-10XUASFRT >STOP > FRT-myr::smGFP-HA in VK00005,pJFRC240- 10XUAS- FRT>STOP > FRT-myr::smGFP-V5-THS-10XUASFRT >STOP > FRT-myr::smGFP-FLAG in su(Hw)attP1) | *Nern et al., 2015* | RRID:BDSC_64088 (Janelia stock 1116898) | |
| Genetic reagent (*Drosophila melanogaster*) | MCFO-5; 57C10PEST_attp8_ 3stop1; (w, R57C10- Flp2::PEST in su(Hw)attP8;; pJFRC201- 10XUAS- FRT>STOP > FRT-myr::smGFPHA in VK00005, pJFRC240-10XUAS-FRT>STOP > FRTmyr::smGFP- V5-THS-10XUAS-FRT>STOP > FRTmyr::smGFP- FLAG in su(Hw)attP1/TM2) | *Nern et al., 2015* | RRID:BDSC_64089 (Janelia stock 1116876) | |
| Genetic reagent (*Drosophila melanogaster*) | MCFO-6; 57C10L_attp8_ 4stop1; (w, R57C10-FlpL in su(Hw)attp8;; pJFRC210-10XUASFRT >STOP > FRT- myr::smGFP-OLLAS in attP2, pJFRC201- 10XUAS- FRT>STOP > FRT-myr::smGFP-HA in VK00005, pJFRC240-10XUAS-FRT>STOP > FRT-myr::smGFP- V5-THS10XUAS-FRT>STOP > FRT-myr::smGFPFLAG in su(Hw)attP1/TM2) | *Nern et al., 2015* | RRID:BDSC_64090 (Janelia stock 1116894) | |
| Genetic reagent (*Drosophila melanogaster*) | MCFO-7; 57C10PEST_attp18_ 4stop1; (w, R57C10- Flp2::PEST in attp18;; pJFRC210-10XUASFRT >STOP > FRT-myr::smGFP-OLLAS in attP2, pJFRC201- 10XUAS-FRT>STOP > FRT-myr::smGFPHA in VK00005, pJFRC240-10XUAS-FRT>STOP > FRTmyr::smGFP-V5-THS-10XUAS-FRT>STOP > FRTmyr::smGFP-FLAG in su(Hw)attP1/TM2) | *Nern et al., 2015* | RRID:BDSC_64091 (Janelia stock 1116875) | |
| Genetic reagent (*Drosophila melanogaster*) | MCFO-3 derivative; 57C10L_brp_SNAP_ MCFO_X23_0117; (w, R57C10-FlpL in su(Hw)attP8; brp::Snap / CyO; pJFRC201-10XUAS-FRT>STOP > FRT-myr::smGFPHA in VK00005,pJFRC240-10XUAS- FRT>STOP > FRTmyr::smGFP-V5-THS-10XUAS- FRT>STOP > FRTmyr::smGFP-FLAG in su(Hw)attP1/TM6B) | *Nern et al., 2015*; *Kohl et al., 2014* | RRID:BDSC_64087 (Janelia stock 3023700) | |
| Genetic reagent (*Drosophila melanogaster*) | 57C10PEST_brp_SNAP_ MCFO_X23_0099; (w, R57C10- Flp2::PEST in attP18; brp::Snap / CyO; pJFRC201-10XUASFRT >STOP > FRT-myr::smGFP-HA in VK00005,pJFRC240- 10XUAS-FRT>STOP > FRT-myr::smGFP-V5-THS-10XUASFRT >STOP > FRT- myr::smGFP-FLAG in su(Hw)attP1/TM6B) | *Nern et al., 2015* | (Janelia stock 3023701) | |
| Genetic reagent (*Drosophila melanogaster*) | MCFO-1 derivative; pBPhsFlp2_PEST_ brp_SNAP_ MCFO_0128; (w, pBPhsFlp2::PEST in attP3; brp::Snap / CyO; pJFRC201- 10XUAS-FRT>STOP > FRT-myr::smGFPHA in VK00005,pJFRC240-10XUAS- FRT>STOP > FRTmyr::smGFP-V5-THS-10XUAS- FRT>STOP > FRTmyr::smGFP-FLAG in su(Hw)attP1/TM6B) | *Nern et al., 2015*; *Kohl et al., 2014* | RRID:BDSC_64085 (Janelia stock 3023951) | |
| Genetic reagent (*Drosophila melanogaster*) | pJFRC2-10XUAS-IVS-mCD8::GFP | *Pfeiffer et al., 2010* | RRID:BDSC_32185 (Janelia stock 1115125) | |
| Genetic reagent (*Drosophila melanogaster*) | UAS_Chrimson_Venus_X_0070; (20XUAS-CsChrimson-mVenus trafficked in attP18) | *Klapoetke et al., 2014* | RRID:BDSC_55134 (Janelia stock 1150416) | |
| Genetic reagent (*Drosophila melanogaster*) | UAS_CsChrimson_Venus_X_0107; (UAS-Syt-HA, 20XUAS-CsChrimson-mVenus trafficked in attP18) | *Klapoetke et al., 2014*; *Robinson et al., 2002* | (Janelia stock 3028408) | |
| Genetic reagent (*Drosophila melanogaster*) | hsPESTOPT_attP3_3stop1_X_0036; (MCFO-1; (w, pBPhsFlp2::PEST in attP3;; pJFRC201- 10XUASFRT >STOP > FRT-myr::smGFP-HA in VK00005,pJFRC240- 10XUAS-FRT>STOP > FRT- myr::smGFP-V5-THS-10XUASFRT >STOP > FRT- myr::smGFPFLAG in su(Hw)attP1/TM3,Sb)) | *Nern et al., 2015* | RRID:BDSC_64085 (Janelia stock 1117734) | |
| Genetic reagent (*Drosophila melanogaster*) | 57C10L_brp_SNAP_MCFO_X23_0117; (MCFO-3 derivative; (w, R57C10-FlpL in su(Hw)attP8; brp::Snap / CyO; pJFRC201-10XUAS-FRT>STOP > FRT-myr::smGFPHA in VK00005,pJFRC240-10XUAS- FRT>STOP > FRTmyr::smGFP-V5-THS-10XUAS- FRT>STOP > FRTmyr::smGFP-FLAG in su(Hw)attP1/TM6B)) | *Kohl et al., 2014*; *Nern et al., 2015* | (Janelia stock 3023700) | |
| Genetic reagent (*Drosophila melanogaster*) | 57C10PEST_attp8_3stop1; (MCFO-5; (w, R57C10- Flp2::PEST in su(Hw)attP8;; pJFRC201- 10XUAS- FRT>STOP > FRT-myr::smGFPHA in VK00005, pJFRC240-10XUAS-FRT>STOP > FRTmyr::smGFP- V5-THS-10XUAS-FRT>STOP > FRTmyr::smGFP- FLAG in su(Hw)attP1/TM2)) | *Nern et al., 2015* | RRID:BDSC_64089 (Janelia stock 1116876) | |

| Reagent type (species) or resource | Designation | Source or reference | Identifiers | Additional information |
|---|---|---|---|---|
| Genetic reagent (*Drosophila melanogaster*) | 57C10L_attp8_4stop1; (MCFO-6; (w, R57C10-FlpL in su(Hw)attp8;; pJFRC210-10XUASFRT >STOP > FRT- myr::smGFP-OLLAS in attP2, pJFRC201- 10XUAS- FRT>STOP > FRT-myr::smGFP-HA in VK00005, pJFRC240-10XUAS-FRT>STOP > FRT-myr::smGFP- V5-THS10XUAS-FRT>STOP > FRT-myr::smGFPFLAG in su(Hw)attP1/TM2)) | *Nern et al., 2015* | RRID:BDSC_64090 (Janelia stock 1116894) | |
| Genetic reagent (*Drosophila melanogaster*) | 57C10PEST_attp18_4stop1; (MCFO-7; (w, R57C10- Flp2::PEST in attp18;; pJFRC210-10XUASFRT >STOP > FRT-myr::smGFP-OLLAS in attP2, pJFRC201- 10XUAS-FRT>STOP > FRT-myr::smGFPHA in VK00005, pJFRC240-10XUAS-FRT>STOP > FRTmyr::smGFP-V5-THS-10XUAS-FRT>STOP > FRTmyr::smGFP-FLAG in su(Hw)attP1/TM2)) | *Nern et al., 2015* | RRID:BDSC_64091 (Janelia stock 1116875) | |
| Genetic reagent (*Drosophila melanogaster*) | LexAop_IVS-myr_3_0008; (pJFRC19-13XLexAop2-IVS-myr::GFP in attP2) | *Pfeiffer et al., 2012* | RRID:BDSC_32209 (Janelia stock 1116736) | |
| Genetic reagent (*Drosophila melanogaster*) | UAS_IVS-mCD8_3_0007; (pJFRC2-10XUAS-IVS-mCD8::GFP) | *Pfeiffer et al., 2010* | RRID:BDSC_32185 (Janelia stock 1115125) | |
| Genetic reagent (*Drosophila melanogaster*) | HAV5_X_0083; pJFRC200-10XUAS-IVS-myr::smGFP-HA (attP18), pJFRC216-13XLexAop2-IVS-myr::smGFP-V5 (su(Hw)attP8; Dr e /TM6B) | *Nern et al., 2015* | (Janelia stock 3020172) | |
| Genetic reagent (*Drosophila melanogaster*) | IVS-myr-FLAG_Syt-HA_3_0055; (w;;5XUAS-IVS-myr::smFLAG in VK00005, pJFRC51-3XUAS-IVS-Syt::smHA in su(Hw)attP1) | *Aso et al., 2014* | (Janelia stock 3001064) | |
| Genetic reagent (*Drosophila melanogaster*) | UAS_IVS_myr_3_0009; (pJFRC12-10XUAS-IVS-myr::GFP in attP2) | *Pfeiffer et al., 2010* | (Janelia stock 1115116) | |
| Genetic reagent (*Drosophila melanogaster*) | TLN-V5_myr-FLAG_Syt-HA_23_0037; (w; 3XpJFRC-TLN-smV5 in su(Hw)attP5; 5XUAS-IVS-myr::smFLAG in VK00005_pJFRC51-3XUAS-IVS-Syt::smHA in su(Hw)attP1/CyO::TM6b) | *Nern et al., 2015* | (Janelia stock 2600002) | |
| Genetic reagent (*Drosophila melanogaster*) | TLN-V5_myr-FLAG_Syt-HA_23_0038; (w; 3XpJFRC-TLN-smV5 in su(Hw)attP5; 5XUAS-IVS-myr::smFLAG in VK00005_pJFRC51-3XUAS-IVS-Syt::smHA in su(Hw)attP2/CyO::TM6b) | *Nern et al., 2015* | (Janelia stock 2600003) | |
| Genetic reagent (*Drosophila melanogaster*) | UAS_IVS_mCD8_3_0045; (pJFRC2-10XUAS-IVS-mCD8::GFP in VK00005) | *Pfeiffer et al., 2010* | (Janelia stock 1115339) | |
| Genetic reagent (*Drosophila melanogaster*) | IVS_myr_GFP_X_0072; (pJFRC200-10XUAS-IVS-myr::smGFP-HA in attP18) | *Nern et al., 2015* | RRID:BDSC_62145 (Janelia stock 1116624) | |
| Genetic reagent (*Drosophila melanogaster*) | UAS_IVS_mCD8_3_0073; (pJFRC7-20XUAS-IVS-mCD8::GFP in attP2) | *Pfeiffer et al., 2010* | RRID:BDSC_32194 (Janelia stock 1115387) | |
| Genetic reagent (*Drosophila melanogaster*) | LexAop2_Chrimson_Venus_X_0082; (13XLexAop2-CsChrimson-mVenus trafficked in attP18) | *Klapoetke et al., 2014* | (Janelia stock 1150410) | |
| Genetic reagent (*Drosophila melanogaster*) | 57C10L_suHwattP8_HAV5_FLAG_0098; (R57C10-FlpL in su(Hw)attP8;; pJFRC201-10XUAS-FRT>STOP > FRT-myr::smGFP-HA in VK0005, pJFRC240-10XUAS-FRT>STOP > FRT-myr::smGFP-V5-THS-10XUAS-FRT>STOP > FRT-myr::smGFP-FLAG in su(Hw)attP1/TM2) | *Nern et al., 2015* | RRID:BDSC_64087 (Janelia stock 3022016) | |
| Antibody | Anti-Brp mouse monoclonal nc82 | Developmental Studies Hybridoma Bank (DSHB) | RRID:AB_2314866 | 1:30 |
| Antibody | Anti-HA rabbit monoclonal C29F4 | Cell Signaling Technologies: 3724S | RRID:AB_1549585 | 1:300 |
| Antibody | Anti-FLAG rat monoclonal DYKDDDDK Epitope Tag Antibody | Novus Biologicals: NBP1-06712 | RRID:AB_1625981 | 1:200 |
| Antibody | DyLight 550 conjugated anti-V5 mouse monoclonal | AbD Serotec: MCA1360D550GA | RRID:AB_2687576 | 1:500 |
| Antibody | Anti-rat IgG (H&L) Goat Polyclonal Antibody ATTO 647N Conjugated | Rockland: 612-156-120 | RRID:AB_10893386 | 1:300 |
| Antibody | Alexa Fluor 594 AffiniPure Donkey Polyclonal Anti-Rabbit IgG (H+L) | Jackson ImmunoResearch Labs: 711-585-152 | RRID:AB_2340621 | 1:500 |
| Antibody | Anti-Green Fluorescent Protein (GFP) Rabbit Polyclonal Antibody, Unconjugated | Thermo Fisher Scientific: A-11122 | RRID:AB_221569 | 1:1000 |

| Reagent type (species) or resource | Designation | Source or reference | Identifiers | Additional information |
|---|---|---|---|---|
| Antibody | Goat Polyclonal anti-Rabbit IgG (H+L) Highly Cross- Adsorbed Antibody, Alexa Fluor 488 | Thermo Fisher Scientific: A-11034 | RRID:AB_2576217 | 1:800 |
| Antibody | Goat Polyclonal anti-Mouse IgG (H+L) Highly Cross- Adsorbed Antibody, Alexa Fluor 568 | Thermo Fisher Scientific: A-11031 | RRID:AB_144696 | 1:800 |
| Antibody | Anti-HA High Affinity; Rat monoclonal antibody (clone 3F10) | Roche: 11867423001 | RRID:AB_390918 | 1:100 |
| Antibody | Cy2-AffiniPure Goat Anti-Mouse IgG (H+L) (min X Hu,Bov,Hrs,Rb,Rat Sr Prot) | Jackson ImmunoResearch Labs: 115-225-166 | RRID:AB_2338746 | 1:600 |
| Antibody | Cy3-AffiniPure Goat Anti-Rabbit IgG (H+L) (min X Hu,Ms,Rat Sr Prot) | Jackson ImmunoResearch Labs: 111-165-144 | RRID:AB_2338006 | 1:1000 |
| Software, algorithm | Janelia Workstation | *Rokicki et al., 2025* | | |
| Software, algorithm | NeuronBridge codebase | *Clements et al., 2024*; *Clements et al., 2021* | | |
| Software, algorithm | Fiji | *Schindelin et al., 2012*; https://fiji.sc | RRID:SCR_0022852 | |
| Software, algorithm | Affinity Designer | https://affinity.serif.com/designer/ | RRID:SCR_016952 | |

## Fly driver stocks

Included split-GAL4 driver stocks and references are listed in *Figure 1—source data 1*. See *Table 2* for counts of driver stocks available from Bloomington *Drosophila* Stock Center.

## Fly reporter stocks

Rescreening of the cell-type collection made use of a 20XUAS-CsChrimson-mVenus in attP18 reporter (*Klapoetke et al., 2014*). Additional reporters are listed in the Key Resources Table and *Figure 1—source data 3*.

## Fly crosses and dissection

Flies were raised on standard cornmeal molasses food typically between 21 and 25°C and 50% humidity. Crosses with hs-Flp MCFO were at 21–22°C, except for a 10- to 60-min 37°C heat shock. Other crosses were typically at 25°C for most of development and 22°C in preparation for dissection. Flies were typically dissected at 1–5 days old for the cell-type collection rescreening, 1–8 days old for other non-MCFO crosses and 3–8 days old for MCFO images reported in this study. Dissection and fixation were carried out as previously described (*Aso et al., 2014*; *Nern et al., 2015*). Protocols are available at https://www.janelia.org/project-team/flylight/protocols.

## Immunohistochemistry, imaging, and image processing

Immunohistochemistry, imaging, and image processing were carried out as previously described (*Aso et al., 2014*; *Nern et al., 2015*; *Sterne et al., 2021*; *Meissner et al., 2023*). See Key Resources Table for antibodies. Additional details of image processing, including source code for processing scripts, are available at https://data.janelia.org/pipeline. See *Supplementary file 2* for a user guide to interpreting the image data.

## Cell-type collection rescreening

If inconsistencies were observed in expression between rescreened images and older image data for a line, the cross was repeated. If still inconsistent, the other copy of the stock (all are kept in duplicate) was examined. If only one of the two stock copies gave the expected expression pattern, it was used to replace the other copy. Repeated problems with expression pattern, except for stochasticity within a cell type, were grounds for removal from the collection. If sex differences were observed, crosses were also repeated. Lines reported with sex differences showed the same difference at least twice. We could not eliminate the possibility of stock issues appearing as sex differences by coincidentally appearing different between sexes and consistent within them.

## Line quality levels

Cell-type lines were qualitatively scored first by visual examination of averaged color depth Maximum Intensity Projection (MIP) CNS images (similar to *Supplementary file 1*; *Otsuna et al., 2018*), and then by a second observer examining individual images. Scores were applied as described in *Figure 2*. As images were taken from a variety of reporters, an effort was made to discount background specific to the reporters, especially 5XUAS-IVS-myr::smFLAG in VK00005.

## Line distribution analysis

3029 cell-type split-GAL4 lines were analyzed, each including images of one male and one female fly sample from rescreening. 6058 brains and 6058 VNCs were included. Images were aligned to the JRC2018 Unisex template (*Bogovic et al., 2020*) and segmented using Direction-Selective Local Thresholding (DSLT; *Kawase et al., 2015*). DSLT weight value, neuron volume percentage, MIP ratio against the aligned template, and MIP-weight ratio were adjusted to refine segmentation. DSLT connected fragmented neurons with up to a 5-μm gap, enhancing neuronal continuity. Noise was eliminated and objects were separated into individual neurons using a connecting component algorithm. The algorithm distinguishes individual neurons by recognizing connected voxel clusters as distinct entities based on DSLT 3D segmentation, followed by removal of voxels below 45/4095 (12 bit) brightness and objects smaller than 30 voxels.

The segmentation produced 19,313 non-overlapping brain objects and 18,860 VNC objects, consisting of neurons, trachea, debris, and antibody background. Most non-neuronal objects and some very dim neurons were eliminated using color depth MIP-based 2D shape filters and machine learning filters, followed by visual inspection, yielding 10,778 brain and 9310 VNC objects. We observed 74 objects containing tissue background with neurons. The background was manually 3D removed using VVDViewer. Neuron masks were combined into a single volume heat map with a newly developed Fiji plugin 'Seg_volume_combine_to_heatmap.jar' (https://github.com/JaneliaSciComp/3D-fiber-auto-segmentation/tree/main; *Otsuna, 2023*). Voxel brightness indicates the number of objects (at most one per line) within each voxel.

## Acknowledgements

This work is part of the FlyLight Project Team at Janelia Research Campus, Howard Hughes Medical Institute, Ashburn, VA. Author order includes the following alphabetical groups: FlyLight Project Team, Janelia Fly Facility, Janelia Scientific Computing Shared Resource, and contributing laboratories. We thank Kristin Scott for visitor project mentorship and the generation of lines labeling the subesophageal ganglion. We thank Arnim Jenett and Teri Ngo for early contributions to split-GAL4 screening. We thank Yichun Shuai for annotations of lines generated with the Aso lab. We thank Project Technical Resources for management coordination and staff support. We thank Melanie Radcliff and Crystal Sullivan Di Pietro for administrative support. We thank Mark Bolstad, Tom Dolafi, Leslie L Foster, Sean Murphy, Donald J Olbris, Todd Safford, Eric Trautman, and Yang Yu for their work on software infrastructure. We thank Olivia DeThomasis, Raj Nair, Hina Naz, Jessalyn Pugh, Brian Rebhorn, Rose Rogall, and Rodney Simmons for preparation of fly food. We thank David Shepherd and Sam Whitehead for their help with analyzing VNC interneuron types. We thank Stephan Saalfeld for imaging discussions. Stocks obtained from the Bloomington *Drosophila* Stock Center (NIH P40OD018537) were used in this study. We thank them, especially Annette Parks, Cale Whitworth, and Sam Zheng, for the maintenance and distribution of stocks from the Janelia collection. We thank Rob Court, Alex McLachlan, and David Osumi-Sutherland for their work on Virtual Fly Brain and coordination on image distribution. We thank Brian Crooks and Zbigniew Iwinski of Zeiss for microscopy assistance. This work was supported in part by the Janelia Visiting Scientist Program. This article is subject to HHMI's Open Access to Publications policy. HHMI lab heads and project team leads have previously granted a nonexclusive CC BY 4.0 license to the public and a sublicensable license to HHMI in their research articles. Pursuant to those licenses, the author-accepted manuscript of this article can be made freely available under a CC BY 4.0 license immediately upon publication.

## Additional information

### Competing interests

Marta Zlatic: Reviewing editor, eLife. The other authors declare that no competing interests exist.

### Funding

| Funder | Grant reference number | Author |
|---|---|---|
| Howard Hughes Medical Institute | | |

The funders had no role in study design, data collection and interpretation, or the decision to submit the work for publication.

### Author contributions

Geoffrey W Meissner, Data curation, Formal analysis, Supervision, Validation, Investigation, Visualization, Methodology, Writing – original draft, Project administration, Writing – review and editing; Allison Vannan, Data curation, Validation, Investigation, Project administration; Jennifer Jeter, Validation, Investigation, Visualization, Project administration; Kari Close, Gina M DePasquale, Kaitlyn Forster, Jaye Anne Beringer, Theresa Gibney, Joanna H Hausenfluck, Yisheng He, Lauren Johnson, Nirmala A Iyer, Hua-Peng Liaw, Scott Miller, Reeham Motaher, Alexandra Novak, Omotara Ogundeyi, Alyson Petruncio, Sophia Protopapas, Jennifer Taylor, Brianna Yarbrough, Heather Dionne, Claire Angstadt, Kelly Ashley, Tam Dang, Guillermo A Gonzalez III, Karen L Hibbard, Cuizhen Huang, Monti Mercer, Brenda Perez, Danielle Ruiz, Viruthika Vallanadu, Grace Zhiyu Zheng, FlyLight Project Team, Investigation; Zachary Dorman, Rachel Lazarus, Kelley Lee, Hsing-Hsi Li, Jacquelyn Price, Susana Tae, Rebecca Vorimo, Validation, Investigation, Project administration; Kristin Henderson, Investigation, Writing – review and editing; Rebecca M Johnston, Validation, Investigation, Methodology, Project administration; Gudrun Ihrke, Supervision, Project administration, Writing – review and editing; Brian Melton, Kevin Xiankun Zeng, Jui-Chun Kao, Scarlett Rose Pitts, Validation, Investigation; Christopher T Zugates, Supervision, Investigation, Project administration; Amanda Cavallaro, Supervision, Validation, Investigation, Project administration; Todd Laverty, Oz Malkesman, Supervision, Project administration; Cristian Goina, Software; Hideo Otsuna, Data curation, Software, Formal analysis, Visualization, Methodology; Konrad Rokicki, Software, Supervision, Visualization, Writing – review and editing; Robert R Svirskas, Software, Visualization; Han SJ Cheong, Michael-John Dolan, Erica Ehrhardt, Kai Feng, Basel El Galfi, Stephen J Huston, Nan Hu, Masayoshi Ito, Claire McKellar, Ryo Minegishi, Shigehiro Namiki, Catherine E Schretter, Lalanti Venkatasubramanian, Kaiyu Wang, Ming Wu, Data curation, Investigation; Jens Goldammer, Gabriella R Sterne, Tanya Wolff, Data curation, Investigation, Writing – review and editing; Aljoscha Nern, Conceptualization, Data curation, Investigation, Methodology, Writing – review and editing; Reed George, Supervision; Yoshinori Aso, Conceptualization, Data curation, Supervision, Investigation, Methodology, Writing – review and editing; Gwyneth M Card, Barry J Dickson, James W Truman, Conceptualization, Supervision; Wyatt Korff, Conceptualization, Supervision, Funding acquisition; Kei Ito, Marta Zlatic, Conceptualization, Supervision, Writing – review and editing; Gerald M Rubin, Conceptualization, Supervision, Funding acquisition, Investigation, Methodology, Writing – original draft, Project administration, Writing – review and editing

### Author ORCIDs

Geoffrey W Meissner ⓘ https://orcid.org/0000-0003-0369-9788
Zachary Dorman ⓘ https://orcid.org/0000-0001-9933-7217
Theresa Gibney ⓘ https://orcid.org/0000-0001-5461-724X
Kristin Henderson ⓘ https://orcid.org/0000-0002-9265-7709
Karen L Hibbard ⓘ https://orcid.org/0000-0002-2001-6099
Cristian Goina ⓘ https://orcid.org/0000-0003-2835-7602
Hideo Otsuna ⓘ https://orcid.org/0000-0002-2107-8881
Konrad Rokicki ⓘ https://orcid.org/0000-0002-2799-9833
Robert R Svirskas ⓘ https://orcid.org/0000-0001-8374-6008
Erica Ehrhardt ⓘ https://orcid.org/0000-0002-9252-1414
Catherine E Schretter ⓘ https://orcid.org/0000-0002-3957-6838
Ming Wu ⓘ https://orcid.org/0000-0002-2193-8271

Yoshinori Aso https://orcid.org/0000-0002-2939-1688
Gwyneth M Card https://orcid.org/0000-0002-7679-3639
Barry J Dickson https://orcid.org/0000-0003-0715-892X
Wyatt Korff https://orcid.org/0000-0001-8396-1533
Kei Ito https://orcid.org/0000-0002-7274-5533
James W Truman https://orcid.org/0000-0002-9209-5435
Marta Zlatic https://orcid.org/0000-0002-3149-2250
Gerald M Rubin https://orcid.org/0000-0001-8762-8703

Reviewer #1 (Public review): https://doi.org/10.7554/eLife.98405.3.sa1
Reviewer #2 (Public review): https://doi.org/10.7554/eLife.98405.3.sa2
Author response https://doi.org/10.7554/eLife.98405.3.sa3

## Additional files

### Supplementary files
Supplementary file 1. Images of cell-type-specific adult split-GAL4 lines. Images are averaged color depth MIPs from rescreened and raw collection SS screen and polarity neuron channel images (*Otsuna et al., 2018*). Images were inverted, overlaid on a 2D outline of JRC2018, and composited. Depth color scale for inverted color depth MIPs is on first page, running from yellow on the anterior brain (or ventral VNC) to blue on the posterior brain (or dorsal VNC).

Supplementary file 2. Guide to FlyLight data. A guide to interpreting the images at https://gen1mcfo.janelia.org and https://splitgal4.janelia.org further describes data organization, labeling, and imaging methods. Images at https://www.janelia.org/gal4-gen1 are processed as described in *Jenett et al., 2012*.

MDAR checklist

### Data availability
The footprint of this image resource (~200 TB) exceeds our known current practical limits on standard public data repositories. Thus, we have made all the primary adult data (and a variety of processed outputs) used in this study freely available under a CC BY 4.0 license through addition to https://doi.org/10.25378/janelia.21266625.v1 and the publicly accessible websites https://splitgal4.janelia.org and https://flylight-raw.janelia.org. Many images are being made searchable with the same permissions on the user-friendly NeuronBridge website https://neuronbridge.janelia.org. NeuronBridge code is available at *Clements et al., 2021* and the application and implementation are discussed further in *Clements et al., 2024*. Larval images are available at https://raw.larval.flylight.virtualflybrain.org/. All other data generated or analyzed during this study are included in the manuscript and supporting files.

The following previously published dataset was used:

| Author(s) | Year | Dataset title | Dataset URL | Database and Identifier |
|---|---|---|---|---|
| Meissner G, Nern A, Dorman Z, DePasquale GM, Forster K, Gibney T, et al | 2022 | Fly Brain Anatomy: FlyLight Gen1 and Split-GAL4 Imagery | https://doi.org/10.25378/janelia.21266625.v1 | Janelia Research Campus, 10.25378/janelia.21266625.v1 |

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
