## [Editor Report · eLife Assessment]

This **valuable** study presents a resource for researchers using *Drosophila* to study neural circuits, in the form of a collection of split-Gal4 lines with an online search engine, which will facilitate the mapping of neuronal circuits. The evidence is **convincing** to demonstrate the utility of these new tools, and of the search engine, for understanding expression patterns in adults and larvae, and differences between the sexes. These resources will be of broad interest to Drosophila researchers in the field of neurobiology.

---

## [Referee Report · Reviewer #1 (Public review)]

Summary:

Meissner et al describe an update on the collection of split-GAL4 lines generated by a consortium led by Janelia Research Campus. This follows the same experimental pipeline described before and presents as a significant increment to the present collection. This will strengthen the usefulness and relevance of "splits" as a standard tool for labs that already use this tool and attract more labs and researchers to use it.

Strengths:

This manuscript presents a solid step to establish Split-GAL4 lines as a relevant tool in the powerful *Drosophila* toolkit. Not only the raw number of available lines contribute to the relevance of this tool in the "technical landscape" of genetic tools, but additional features of this effort contribute to the successful adoption. These include:

(1) A description of expression patterns in the adult and larvae, expanding the "audience" for these tools

(2) A classification of line combination according to quality levels, which provides a relevant criterion while deciding to use a particular set of "splits".

(3) Discrimination between male and female expression patterns, providing hints regarding the potential role of these gender-specific circuits.

(4) The search engine seems to be user-friendly, facilitating the retrieval of useful information.

(5) An acknowledgement of the caveats and challenges that splits (like any other genetic tool) can carry.

Overall, the authors employed a pipeline that maximizes the potential of the Split-GAL4 collection to the scientific community.

Weaknesses:

My concerns were resolved regarding the existence of caveats while using these tools that researchers should be aware of, particularly those using them for the first time.

---

## [Referee Report · Reviewer #2 (Public review)]

Summary:

This manuscript describes the creation and curation of a collection of genetic driver lines that specifically label small numbers of neurons, often just a single to handful of cell types, in the central nervous system of the fruit fly, *Drosophila melanogaster*. The authors screened over 77,000 split hemidriver combinations to yield a collection of 3060 lines targeting a range of cell types in the adult Drosophila central nervous system and 1373 lines characterized in third-instar larvae. These genetic driver lines have already contributed to several important publications and will no doubt continue to do so. It is a truly valuable resource that represents the cooperation of several labs throughout the Drosophila community.

Strengths:

The authors have thoughtfully curated and documented the lines that they have created, so that they may be maximally useful to the greater community. This documentation includes confocal images of neurons labeled by each driver line and when possible, a list of cell types labeled by the genetic driver line and their identity in an EM connectome dataset. The authors have also made available some information from the other lines they created and tested but deemed not specific or strong enough to be included as part of the collection. This additional resource will be a valuable aid for those seeking to label cell types that may not be included in the main collection.

The added revisions help to clarify important points relating to the creation of the lines, which lines were included as part of this specific collection, and caveats to be mindful of when using any of the described lines. These revisions will increase the manuscript's utility to users who may be less familiar with this resource.

Weaknesses:

The major weakness, which is also in some ways a strength, is the stringent requirement that lines that be included be highly specific across the CNS. As a result, the lines that are part of this specific collection are sparse and specific but also limited in which cell types they cover. Doubtless there are many missing cell types.

---

## [Author Response]

The following is the authors’ response to the original reviews.

**Public Reviews:**

**Reviewer #1 (Public Review):**
Summary:Meissner et al describe an update on the collection of split-GAL4 lines generated by a consortium led by Janelia Research Campus. This follows the same experimental pipeline described before and presents as a significant increment to the present collection. This will strengthen the usefulness and relevance of "splits" as a standard tool for labs that already use this tool and attract more labs and researchers to use it.Strengths:This manuscript presents a solid step to establish Split-GAL4 lines as a relevant tool in the powerful *Drosophila* toolkit. Not only does the raw number of available lines contribute to the relevance of this tool in the "technical landscape" of genetic tools, but additional features of this effort contribute to the successful adoption. These include:(1) A description of expression patterns in the adult and larvae, expanding the "audience" for these tools(2) A classification of line combination according to quality levels, which provides a relevant criterion while deciding to use a particular set of "splits".(3) Discrimination between male and female expression patterns, providing hints regarding the potential role of these gender-specific circuits.(4) The search engine seems to be user-friendly, facilitating the retrieval of useful information.Overall, the authors employed a pipeline that maximizes the potential of the Split-GAL4 collection to the scientific community.Weaknesses:The following aspects apply:The use of split-GAL4 lines has improved tremendously the genetic toolkit of *Drosophila* and this manuscript is another step forward in establishing this tool in the genetic repertoire that laboratories use. Thus, this would be a perfect opportunity for the authors to review the current status of this tool, addressing its caveats and how to effectively implement it into the experimental pipeline.(1) While the authors do bring up a series of relevant caveats that the community should be aware of while using split-GAL4 lines, the authors should take the opportunity to address some of the genetic issues that frequently arise while using the described genetic tools. This is particularly important for laboratories that lack the experience using split-GAL4 lines and wish to use them. Some of these issues are covertly brought up, but not entirely clarified.First, why do the authors (wisely) rescreen the lines using UAS-CsChrimson-mVenus? One reason is that using another transgene (such as UAS-GFP) and/or another genomic locus can drive a different expression pattern or intensities. Although this is discussed, this should be made more explicit and the readers should be aware of this.Second, it would be important to include a discussion regarding the potential of hemidriver lines to suffer from transvection effects whenever there is a genetic element in the same locus. These are serious issues that prevent a more reliable use of split-GAL4 lines that, once again, should be discussed.

We added additional explanatory text to the discussion.

(2) The authors simply mention that the goal of the manuscript is to "summarize the results obtained over the past decade.". A better explanation would be welcomed in order to understand the need of a dedicated manuscript to announce the availability of a new batch of lines when previous publications already described the Split-GAL4 lines. At the extreme, one might question why we need a manuscript for this when a simple footnote on Janelia's website would suffice.

We added an additional mention of the cell type split-GAL4 collection at the relevant section and added more emphasis on the curation process adding value to the final selections. We feel that the manuscript is useful to document the methods used for the contained analysis and datasets and gives a starting point to the reader to go through the many split-GAL4 publications and images.

**Reviewer #2 (Public Review):**
Summary: This manuscript describes the creation and curation of a collection of genetic driver lines that specifically label small numbers of neurons, often just a single to handful of cell types, in the central nervous system of the fruit fly, *Drosophila melanogaster*. The authors screened over 77,000 split hemidriver combinations to yield a collection of 3060 lines targeting a range of cell types in the adult Drosophila central nervous system and 1373 lines characterized in third-instar larvae. These genetic driver lines have already contributed to several important publications and will no doubt continue to do so. It is a truly valuable resource that represents the cooperation of several labs throughout the Drosophila community.Strengths:The authors have thoughtfully curated and documented the lines that they have created, so that they may be maximally useful to the greater community. This documentation includes confocal images of neurons labeled by each driver line and when possible, a list of cell types labeled by the genetic driver line and their identity in an EM connectome dataset. The authors have also made available some information from the other lines they created and tested but deemed not specific or strong enough to be included as part of the collection. This additional resource will be a valuable aid for those seeking to label cell types that may not be included in the main collection.Weaknesses:None, this is a valuable set of tools that took many years of effort by several labs. This collection will continue to facilitate important science for years to come.

We thank the reviewer for their positive feedback.

**Reviewer #3 (Public Review):**
Summary:The manuscript by Meissner et al. describes a collection of 3060 *Drosophila* lines that can be used to genetically target very small numbers of brain cells. The collection is the product of over a decade of work by the FlyLight Project Team at the Janelia Research Campus and their collaborators. This painstaking work has used the intersectional split-Gal4 method to combine pairs of so-called hemidrivers into driver lines capable of highly refined expression, often targeting single cell types. Roughly one-third of the lines have been described and characterized in previous publications and others will be described in manuscripts still in preparation. They are brought together here with many new lines to form one high-quality collection of lines with exceptional selectivity of expression. As detailed in the manuscript, all of the lines described have been made publicly available accompanied by an online database of images and metadata that allow researchers to identify lines containing neurons of interest to them. Collectively, the lines include neurons in most regions of both the adult and larval nervous systems, and the imaging database is intended to eventually permit anatomical searching that can match cell types targeted by the lines to those identified at the EM level in emerging connectomes. In addition, the manuscript introduces a second, freely accessible database of raw imaging data for many lower quality, but still potentially useful, split-Gal4 driver lines made by the FlyLight Project Team.Strengths:Both the stock collection and the image databases are substantial and important resources that will be of obvious interest to neuroscientists conducting research in *Drosophila*. Although many researchers will already be aware of the basic resources generated at Janelia, the comprehensive description provided in this manuscript represents a useful summary of past and recent accomplishments of the FlyLight Team and their collaborators and will be very valuable to newcomers in the field. In addition, the new lines being made available and the effort to collect all lines that have been generated that have highly specific expression patterns is very useful to all.Weaknesses:The collection of lines presented here is obviously somewhat redundant in including lines from previously published collections. Potentially confusing is the fact that previously published split-Gal4 collections have also touted lines with highly selective expression, but only a fraction of those lines have been chosen for inclusion in the present manuscript. For example, the collection of Shuai et al. (2023) describes some 800 new lines, many with specificity for neurons with connectivity to the mushroom body, but only 168 of these lines were selected for inclusion here. This is presumably because of the more stringent criteria applied in selecting the lines described in this manuscript, but it would be useful to spell this out and explain what makes this collection different from those previously published (and those forthcoming).

We added more description of how this collection is focused on the best cell-type-specific lines across the CNS. An important requirement for inclusion was this degree of specificity across the CNS, while many prior publications had a greater emphasis on lines with a narrower focus of specificity.

**Recommendations for the authors:**

**Reviewer #1 (Recommendations For The Authors):**
Luckily for us, genetics is for the most part an exact science. However, there's still some "voodoo" in a lot of genetic combinations that the authors should disclose and be as clear as possible in the manuscript. This allows for the potential users to gauge expectations and devise a priori alternative plans.

We attempted to comprehensively cover the caveats inherent in our genetic targeting approach.

Minor points:(1) The authors mention that fly age should be controlled as expression can vary. Is there any reference to support this claim?

We added a reference describing driver expression changes over development.

(2) There should be a citation for "Flies were typically 1-5 days old at dissection for the cell type collection rescreening, 1-8 days old for other non-MCFO crosses and 3-8 days old for MCFO".

We clarified that these descriptions were of our experimental preparations, not describing other citable work.

**Reviewer #3 (Recommendations For The Authors):**
General Points:Overall, the manuscript is very clear, but there are a couple of points where more explicit information would be useful. One of these is with respect to the issue of selectivity of targeting. The cell type specificity of lines is often referred to, but cell types can range from single pairs of neurons to hundreds of indistinguishable neurons with similar morphology and function. It would be useful if the authors explained whether their use of the term "cell type" distinguishes cell type from cell number. It would also be useful if lines that target many neurons of a single cell type were identified.

We added further discussion of cell types vs. cell numbers. Our labeling strategy was not optimized for counting cell numbers labeled by each line. We believe EM studies are best positioned to comprehensively evaluate the number of cells making up each type.

The second point relates to vagueness about the intended schedule for providing resources that will match (or allow matching of) neurons to the connectome. For example, on pp. 5-6 it is stated that: "In the future all of the neurons in these lines will be uniquely identified and linked to neurons reconstructed in the electron microscopy volume of the larva" but no timeline is provided. Similarly, for the adult neurons it is stated on p. 4 that: "Anatomical searching for comparison to other light microscopy (LM) and EM data is being made available." A more explicit statement about what resources are and are not yet available, a timeline for full availability, and an indication of how many lines currently have been matched to EM data would be helpful.

During the review and revision period we have made progress on processing the images in the collection. We updated the text with the current status and anticipated timeline for completion.

Specific Points:p. 4 "Although the lines used for these comparisons are not a random sample, the areas of greatest difference are in the vicinity of previously described sexual dimorphisms..." In the vicinity of is a very vague statement of localization. A couple of examples of what is meant here would be useful.

We added example images to Figure 3.

p. 5 "...may have specific expression outside our regions of interest." It's not clear what "our regions of interest" refers to here. Please clarify.

We clarified that we were referring to the regions studied in the publications listed in Table 1.

p. 5 "...lines that were sparse in VNC but dirty in the brain or SEZ..." A more quantitative descriptor than "dirty" would be helpful.

We unfortunately did not quantify the extent of undesired brain/SEZ expression, but attempted to clarify the statement.

p. 6 "...the images are being made instantly searchable for LM and EM comparisons at NeuronBridge..." Here again it is hard to know what is meant by "being made instantly searchable." How many have been made searchable and what is the bottleneck in making the rest searchable?

We updated the text as described above. The bottleneck has been available processing capacity for the hundreds of thousands of included images.

Figure 1 Supplemental File 2: The movie is beautiful, but it seems more useful as art than as a reference. Perhaps converting it to a pdf of searchable images for each line would make it more useful.

We replaced the movie with a searchable PDF.

Fig. 2(B) legend: "Other lines may have more than two types." It is not clear what "other lines" are being referred to.

As part of making the quality evaluation more robust, we scored lines for the clear presence of three or more cell types. We updated the text accordingly.

Fig. 2(C): Presumably the image shown is an example of variability in expression rather than weakness, but it is hard to know without a point of comparison. Perhaps show the expression patterns of other samples? Or describe briefly in the legend what other samples looked like?

We added Figure 2 - figure supplement 1 with examples of variable expression in a split-GAL4 line.